# Leveraging the Structure of DNAJA1 to Discover Novel Potential Pancreatic Cancer Therapies

**DOI:** 10.3390/biom12101391

**Published:** 2022-09-29

**Authors:** Heidi E. Roth, Aline De Lima Leite, Nicolas Y. Palermo, Robert Powers

**Affiliations:** 1Department of Chemistry, University of Nebraska-Lincoln, Lincoln, NE 68588, USA; 2Nebraska Center for Integrated Biomolecular Communication, University of Nebraska-Lincoln, Lincoln, NE 68588, USA; 3Computational Chemistry Core Facility, VCR Cores, University of Nebraska Medical Center, Omaha, NE 68198, USA

**Keywords:** DNAJA1, pancreatic cancer, NMR structural biology, drug discovery, Hsp40

## Abstract

Pancreatic cancer remains one of the deadliest forms of cancer with a 5-year survival rate of only 11%. Difficult diagnosis and limited treatment options are the major causes of the poor outcome for pancreatic cancer. The human protein DNAJA1 has been proposed as a potential therapeutic target for pancreatic cancer, but its cellular and biological functions remain unclear. Previous studies have suggested that DNAJA1′s cellular activity may be dependent upon its protein binding partners. To further investigate this assertion, the first 107 amino acid structures of DNAJA1 were solved by NMR, which includes the classical J-domain and its associated linker region that is proposed to be vital to DNAJA1 functionality. The DNAJA1 NMR structure was then used to identify both protein and ligand binding sites and potential binding partners that may suggest the intracellular roles of DNAJA1. Virtual drug screenings followed by NMR and isothermal titration calorimetry identified 5 drug-like compounds that bind to two different sites on DNAJA1. A pull-down assay identified 8 potentially novel protein binding partners of DNAJA1. These proteins in conjunction with our previously published metabolomics study support a vital role for DNAJA1 in cellular oncogenesis and pancreatic cancer.

## 1. Introduction

According to the American Cancer Society, 62,210 people in the USA will be diagnosed with pancreatic cancer in 2022 with 49,830 resulting in fatalities [1]. Pancreatic cancer survival rates have seen minimal improvement in the last thirty years. The stage-one 5-year survival rate is only 11%, with just a few patients being diagnosed at such an early stage of the disease. Early diagnosis of pancreatic cancer is often difficult as the initial stages tend to be asymptomatic and tumor presence is not outwardly apparent [1]. Unfortunately, the 5-year survival rate decreases with stage progression hitting a low of 3% by stage three [1]. These poor outcomes are due to a delayed diagnosis, rapid acquisition of drug resistance [2,3], and limited options for treatment. Developing treatments for pancreatic cancer is an extremely daunting challenge that has enormously profound implications for human health. Identifying a therapeutic target could lead to improvements in disease treatment and improve pancreatic cancer survival rates, which have otherwise remained stagnant.

Towards this end, we describe the discovery of novel, drug-like compounds that bind Human protein DnaJ homolog subfamily A member 1 (DNAJA1) and the identification of potential in vivo protein binding partners of DNAJA1. The solution structure of the first 107 amino acids of DNAJA1 was solved using nuclear magnetic resonance (NMR). The NMR structure of DNAJA1 was then leveraged to aid in identifying chemical leads and the location of their binding sites. A virtual ligand screen employed Molegro Virtual Docker (MVD) and the ChemBridge™ Diversity Library containing 100,000 structures, which identified 5 potential binders to DNAJA1. The chemical leads from the virtual screening were confirmed to bind DNAJA1 by one-dimensional (1D) and two-dimensional (2D) NMR, and by isothermal titration calorimetry (ITC). Protein binding partners of DNAJA1 were also probed via an in vitro pull-down assay, which identified 8 potential protein binding partners to DNAJA1.

DNAJA1 has been identified as a potential therapeutic target for treating pancreatic ductal adenocarcinoma (PDAC) [4,5,6,7], which accounts for more than 90% of all pancreatic cancers [8]. The human DnaJ protein family contains 41 members that make up the heat shock protein 40 (Hsp40) class [9]. This family is characterized by the presence of a J-domain, a highly conserved motif that is responsible for Hsp40 co-chaperone activity [10]. Hsp40s interact with Hsp70s (DnaK) to stimulate ATP hydrolysis and facilitate protein folding [11]. Hsp70s are responsible for several cellular processes such as rescuing misfolded proteins, folding polypeptide chains, transport of polypeptides through membranes, assembly and disassembly of protein complexes, and control of regulatory proteins [9,12,13]. Thus, DNAJA1 is also involved in protein folding, protein clearance, protein translocation [14], and cellular stress response [15]. Specifically, DNAJA1 appears to be involved in importing proteins into the mitochondria [16,17]. The mitochondrial pathway to apoptosis protects against cancer and requires importing apoptotic factors into the mitochondrial membrane [18,19,20,21]. In PDAC, it has been proposed that DNAJA1 may regulate the anti-apoptotic state through the hyperphosphorylation of c-Jun [4]. This mechanism likely involves an unknown protein binding partner of DNAJA1 and is consistent with DNAJA1 being highly networked with proteins associated with pancreatic cancer [4,5,6,7,9,22,23,24,25,26,27,28,29]. For example, DNAJA1 has been observed to stabilize mutant p53 proteins, preventing its degradation, impacting its localization, and allowing p53 to promote metastasis in pancreatic cancer cells. Other heat-shock proteins were shown to be regulators of apoptosis, where DnaK/Hsp70 suppresses JNK activity [30,31,32].

While Hsp40s have been largely assumed to only assist with protein folding, it has been proposed that J-domain proteins may have functions independent of co-chaperoning [33]. A previous observation that DNAJA1 expression was increased in head and neck squamous cell carcinoma in combination with its binding of unfolded, mutant p53 suggests an oncogenic role [7]. The down regulation of DNAJA1 in C6 glioblastoma was shown to increase tumorigenicity with the rescued expression of DNAJA1 reversing this behavior [22]. Conversely, DNAJA1 down regulation in colorectal cancer resulted in an increase in cellular proliferation, invasion, and metastasis [27]. These varying impacts of DNAJA1 regulation suggest a complex, but potentially valuable avenue of protein function and regulation in the Hsp40 family.

Proteomic studies of patients with chronic pancreatitis (CP) and PDAC found that DNAJA1 was down regulated 5-fold in PDAC compared to CP and control groups [34]. This DNAJA1 dysregulation combined with the growing literature support for its role in carcinogenic pathways provided the rationale to probe the effect of DNAJA1 on PDAC cell survival. Increased DNAJA1 expression in PDAC cells resulted in a proto-oncogenic effect and promoted cell growth, survival, and metastasis [6]. The increase in DNAJA1 expression in other cancers piqued further interest in its potential role as a therapeutic target.

## 2. Material and Methods

### 2.1. Protein Expression and Purification

The structural studies focused on the N-terminal amino acids comprising residues 1 to 107 of DNAJA1 (DNAJA1-107), which encompass a previously characterized J-domain (residues 1 to 67) [4] and an uncharacterized linker section (residues 68 to 107). Plasmid HR3099J was received from the Northeast Structural Genomics Consortium (NESG) and codes for ampicillin resistance and the first 107 amino acids of DNAJA1 that is preceded by an N-terminal 6x-histidine tag. DNAJA1-107 was over-expressed in one-liter One Shot™ BL21 (DE3) chemically competent *E. coli* cultures with 100 mg/L ampicillin at 37 °C. For unlabeled DNAJA1-107, *E. coli* cells were grown in Luria Bertani medium. For ^15^N-labeled DNAJA1-107, *E. coli* cells were grown in M9 minimal media containing U-^15^N NH_4_Cl. For ^15^N- and ^13^C-labeled DNAJA1-107, *E. coli* cells were grown in M9 minimal media containing 4 g of U-^13^C glucose and 1 g of U-^15^N NH_4_Cl [35]. Media was agitated at 200 rpm at 37 °C with 1 mM IPTG added at an OD_600_ ≥ 0.6. Cells were harvested after an 18-h incubation period with IPTG at room temperature. Cell lysates were purified using a Co^2+^ affinity column (HisPur™ Cobalt Resin, ThermoFisher Scientific, Rockford, IL, USA) and the protein purity was assessed by SDS-PAGE. The approximate yield of purified DNAJA1-107 was 8–10 mg/L in M9 media. Purified DNAJA1-107 samples were maintained in 20 mM 2-(N-morpholino)ethanesulfonic acid (MES) buffer containing 100 mM NaCl and 5 mM CaCl_2_ at pH 6.5. Recombinant Human ADP/ATP translocase 3 (ANT3) was ordered from CUSABIO^®^, catalog number CSB-EP021520HU1. The ANT3 construct comprised amino acids 232–272 and was maintained in 10 mM Tris-HCl, 1 mM EDTA, and 6% trehalose at pH 8.0. The ANT3 construct included an N-terminal 6xHis-KSI tag to aid purification and solubility.

### 2.2. Virtual Ligand Screens

A library of 17 in-house compounds was prepared to establish ligand efficiency reference scores. These compounds are listed in Appendix A. The library included O-phospho-L-serine, which was previously reported as binding the J-domain of DNAJA1 [4]. Structure-data files (SDF) were cleaned with Molegro [36] by removing water molecules, replacing any missing bonds, and adding hydrogen atoms. 

Virtual ligand affinity screens were performed using Molegro Virtual Docker (MVD, Molexus IVS, Odder, Denmark) [36] on the Crane Cluster (7232 Intel Xeon cores in 452 nodes with 64GB RAM per node) within the Holland Computing Center. Molegro incorporates a cavity prediction algorithm that employs an expanded Van der Waals molecular surface search. The cavity prediction algorithm was used with a confined search space radius and a minimum volume of 10Å of the predicted binding pocket for all virtual screens. A predictive DNAJA1-107 protein data bank (pdb) file generated by Phyre2 [37] software was used as the ligand receptor. The ChemBridge™ Diversity Library (https://chembridge.com/diversity-and-pre-plated-libraries/diversity-libraries/ accessed on 20 September 2022) containing 100,000 compounds was imported, cleaned, and then screened by MVD with a maximum of five poses per ligand for a total of 500,000 conformers [36]. Results were scored using the MolDock Score energy with a grid resolution of 0.30Å. The screening results were sorted based on the calculated ligand efficiency scores. All compounds were ordered from Hit2Lead.com^©^ (ChemBridge Corporation, San Diego, CA). Tanimoto correlation coefficients were calculated using ChemMine Tools [38].

### 2.3. In Vitro Ligand Binding Studies

#### 2.3.1. NMR

To assess compound binding specificity, 40 mM stock solutions were prepared in water or dimethyl sulfoxide (DMSO) depending upon compound solubility. Compounds were added to unlabeled DNAJA1-107 or ^15^N-DNAJA1-107 in the 20 mM MES buffer pH 6.5 at a 10:1 ratio and incubated at room temperature for thirty minutes to aid in ligand solubilization. 1D ^31^P NMR line-broadening experiments were collected on the 17 phosphorus containing compounds (Appendix A). Saturation transfer difference (STD) experiments were collected on the 20 compounds identified as potential binders from the virtual ligand screen. 2D ^1^H-^15^N HSQC spectra were then acquired for ^15^N-DNAJA1-107 in the presence of the potential binders that either exhibited line-broadening or an STD in the 1D NMR experiments. STD experiments (Bruker pulse program stddiffesgp) were acquired with 0.5 mM DNAJA1, 32K points, a spectral width of 11,160 Hz and 128 scans. 1D ^31^P NMR line-broadening experiments (zgpg30) were acquired with 0.5 mM DNAJA1, 64K points, a spectral width of 56,818 Hz and 128 scans. 2D ^1^H-^15^N HSQC (hsqcetf3gpsi) spectra were acquired with 1 mM DNAJA1 and 8 scans using 2K points and a spectral width of 11,160 Hz in the ^1^H dimension and 256 points and a spectral width of 2839 Hz in the ^15^N dimension. STD experiments were acquired at 298K on a 600 MHz Bruker AVANCE spectrometer equipped with a 5 mm TXI (^1^H, ^13^C, ^15^N) probe with XYZ Gradients. All other NMR spectra were collected at 298K on a Bruker Avance III-HD 700 MHz spectrometer equipped with a quadruple resonance QCI-P cryoprobe, a SampleJet sample changer, autotune and match, and IconNMR for high-throughput data collection.

#### 2.3.2. Isothermal Titration Calorimetry

ITC data was collected on a MicroCal by GE Healthcare. Ligand stock solutions were prepared at a concentration of 450 µM and DNAJA1-107 was maintained at a concentration of 20 µM. Ligands were titrated stepwise into the DNAJA1-107 stock solution. Buffer only spectra were collected and subtracted from the ligand spectra to remove any potential enthalpy from binding and dilution. The raw data were fit to a standard binding curve with stoichiometry set as a variable to measure a single dissociation constant (K_D_). 

### 2.4. DNAJA1-107 Structural Characterization

Uniformly ^15^N- and ^13^C- labeled DNAJA1-107 NMR experiments were collected with non-uniform sampling at 20% sparsity using a Poisson-gap schedule [39] at 298K on a 700 MHz Bruker Avance III spectrometer equipped with a 5 mm QCI-P probe with cryogenically cooled carbon and proton channels. Backbone and side-chain assignments were obtained with 2D ^1^H-^15^N HSQC, 2D ^1^H-^13^C HSQC, three-dimensional (3D) CBCA(CO)NH, HNCACB, HNCO, HN(CA)CO, HCCH-COSY, HCCH-TOCSY, HNHA, ^15^N-edited HSQC-NOESY, and ^13^C-edited HSQC-NOESY experiments The NOESY spectra were collected with a 120 ms mixing time. All data were reconstructed using SMILE [40] in NMRPipe [41] using NMRBox [42]. Assignments were completed using CcpNmr Analysis V2 (https://www.ccpn.ac.uk/v2-software/software/analysis accessed on 20 September 2022) [43]. An initial model of DNAJA1-107 based on backbone resonance assignments was generated using the CS-ROSETTA web portal (https://csrosetta.bmrb.io/submit accessed on 20 September 2022) [44,45]. This initial model of DNAJA1-107 was refined with XPLOR-NIH (https://nmr.cit.nih.gov/xplor-nih/ accessed on 20 September 2022) [46,47] version 3.3 with the following restraints: 1143 NOE distance restraints, 23 hydrogen bond distance restraints, 39 ^3^*J*_NHα_ coupling constants, and 176 predicted dihedral angle restraints from TALOS+ [48]. 1000 total structures were generated for DNAJA1-107 using an XPLOR-NIH structure refinement. The 20 lowest energy structures were subjected to water refinement following RECCORD parameters. The average of these 20 water-refined structures was further refined by energy minimization. The 20 water-refined structures were analyzed using the PSVS software suite, which includes common structural validation packages [49]. UCSF Chimera was used for visualization [50]. Chemical shift assignments have been deposited into the BMRB with ID 51532. The coordinates of the water-refined ensemble have been deposited into the PDB with ID 8E2O.

### 2.5. Identification of Protein Binding Partners

#### 2.5.1. Mammalian Cell Cultures

The MIA PaCa-2 PDAC cell line was purchased from the American Type Culture Collection (ATCC) and cultured following ATCC guidelines. Cells were grown to 80% confluency followed by on-plate lysis using Pierce^®^ RIPA buffer with HALT™ protease inhibitor cocktail. Cells were incubated on ice with the RIPA buffer for 5 min followed by centrifugation at 14,000× *g* for 15 min. Clarified lysate was then used for Pierce™ His Protein Interaction Pull-Down Kit.

#### 2.5.2. Pull-Down Assay

Protein binding partners of DNAJA1-107 were identified using the Pierce™ His Protein Interaction Pull-Down Kit from ThermoFisher Scientific. 6x-His tagged DNAJA1-107 was expressed in One Shot™ BL21 (DE3) chemically competent *E. coli* and immobilized on a cobalt resin. A negative control was performed by incubating the cell lysate with only the cobalt resin to identify non-specific binding. In Gel Digestion was performed as previously described [51]. Trypsin-digested samples were analyzed by LC-MS using an Acquity UPLC M-Class and a Xevo G2-XS Quadrupole Time-of-Flight Mass Spectrometer with a 60-minute linear gradient from 100% H_2_O + 0.1% FA to 100% ACN + 0.1% FA using a 0.35 µL/min flow rate. The MS source was a nanoflow ESI with MSE continuum acquisition. Scan time was 0.5 s with a mass range of 50–2000 Da. ProteinLynx Global SERVER™ by Waters was used for data processing. Peptide fragments were matched to proteins using reverse entry searching in the UniProt database [52] using FASTA sequences.

## 3. Results and Discussion

### 3.1. DNAJA1-107 Solution Structure

The backbone resonance assignments were completed for 89 (86%) of the 103 amino acid residues excluding the 10-residue histidine tag and 4 proline residues. 14 amino acids were not assigned, which is likely attributed to the unstructured, flexible nature of the N- and C-terminal loop regions where these residues were located. The complete amino acid sequence for DNAJA1-107 is available in Appendix A. The sequence excludes the N-terminal histidine tag and highlights residues with missing assignments in red font. The backbone and side-chain assignments corresponded to 89 of 107 N, 89 of 103 HN, 94 of 107 Cα, 107 of 122 Hα, 80 of 85 Cβ, 128 of 166 Hβ, 58 of 82 Cγ, 66 of 114 Hγ, 45 of 60 Cδ, 44 of 84 Hδ, 23 of 39 Cε, 26 of 51 Hε, 5 of 17 Cζ, and 13 of 19 Hζ. The solution structure of DNAJA1-107 was calculated with 1143 distance restraints, 176 dihedral angle restraints, 39 ^3^*J*_NHα_ coupling constants, and an initial model generated using CS-ROSETTA [44,45]. For structure generation, XPLOR-NIH was used to calculate 1000 DNAJA1-107 structures, and the resulting 20 lowest-energy structures were used for further refinement in a water-simulated environment (Figure 1A,B).

The water refined 20-structure ensemble did not have any distance violations greater than 0.5 Å or dihedral violations greater than 5°. The NMR data were sufficient to define a good-quality DNAJA1-107 structure as evident by an RMSD of 1.0 ± 0.21 Å for the backbone secondary structures and an RMSD of 1.4 ± 0.19 Å for heavy atoms. A comparison between the CS-ROSETTA homology model used for the structure refinement and the final water-refined average structure of DNAJA1-107 yielded a backbone RMSD of 2.0 Å for the full protein and 1.6 Å for only the secondary structures. Complete structural statistics for DNAJA1-107 are listed in Table 1.

Overall structural quality was evaluated using the PSVS software suite (Table 2). 81 residues (90.5%) were in the most favored regions in the Ramachandran analysis and the remaining 8 residues (9.5%) were in the allowed regions. These 8 residues were located across unstructured, flexible regions of the protein. PROCHECK analysis supports dihedral angle quality with Z-scores of 0.63 for φ, ψ angles and 0.18 for all angles. ProsaII yielded a Z-score of 0.83 for overall model quality, which is within the acceptable scoring range of −10 to 10. Verify3D produced a Z-score of −2.25 and MolProbity had a clash score of −2.64. These scores support a high-quality NMR structure that are consistent with structures deposited in the PDB [49].

### 3.2. DNAJA1-107 and DNAJA1-67 Structure Comparison

The J-domain of DNAJA1 comprises the first 67 residues from the N-terminus, which has been previously solved and comprises 4 α-helices [4]. The J-domain structure from DNAJA1 will be referred to as DNAJA1-67. The NMR solution structure presented herein contains an additional 40 amino acids after the J-domain, which was largely unstructured and was comprised of glycine- and phenylalanine-rich regions. This region will be referred to as the linker region. An overlay of the DNAJA1-67 and DNAJA1-107 structures is presented in Figure 1C. A comparison of the two structures yields a backbone RMSD of 3.4 Å for the first 67 residues. An RMSD of 4.4 Å was obtained when comparing only the secondary structure regions. Only three of the four α-helices were present in the DNAJA1-107 structure. The first 9 residues in DNAJA1-107 were unidentifiable, which resulted in the loss of the first α-helix that was observed in the previously solved DNAJA1-67 structure. This suggests the addition of the linker region dramatically increased the flexibility of the N-terminus. 

Both linker length and sequence composition have been previously shown to play important roles in protein stability [53]. Using various lengths of glycine-rich linkers, one study found that a large, 47 residue linker had low thermodynamic stability in single-chain proteins [53]. The importance of the ratio of glycine to serine and alanine was also demonstrated, with either high or low ratios of glycine resulting in lower protein stability [53]. The 40 residue linker of DNAJA1-107 has 2 Ala, 1 Ser, and 12 Gly residues. Thus, the decreased stability of the DNAJA1-107 N-terminus may be attributed to the high ratio of Gly to Ala and Ser in the linker region.

The second α-helix showed minor deviations from its location in the DNAJA1-67 structure. Conversely, the third and fourth α-helices showed a greater deviation from the previous structure. These structural changes are largely attributed to the presence of the C-terminus linker region that appears to promote a structured domain. This is further evidenced by the peak distribution in the overlay of the 2D ^1^H-^15^N HSQC spectra for DNAJA1-67 and DNAJA1-107 (Figure 1D).

The DNAJA1 linker region was chosen to be included in the construct to aid in ligand binding studies. While linker regions are generally rigid to prevent distinct domains from interacting with each other, glycine-rich linkers are flexible [54]. Importantly, glycine-rich linkers are known to have vital functional roles in protein structures by contributing to protein folding and stability, and by supporting transient and weak protein–protein interactions [54]. Previous studies of the Hsp40 family have suggested that the G/F-linker region immediately following the J-domain is vital for its functions, and these two domains alone are likely sufficient for basic Hsp40 function in vivo [55]. While it was anticipated that this G/F-linker region would be largely coiled, α-helical motifs are present at residues 86 Asp to 91 Phe and 97 Arg to 99 Gln. The expected α-helical and coil motifs would be consistent with the most common secondary structures seen in linker residues [56]. Thus, it was surprising and unexpected to observe a structurally disordered linker region that disrupted the J-domain helical structure.

### 3.3. Virtual Ligand Affinity Screens

To identify drug-like compounds that mitigate the function of DNAJA1, a virtual ligand affinity screen was conducted using two different chemical libraries and MVD [36]. For these initial screens, a Phyre2 homology model of DNAJA1-107 was used in place of the NMR solution structure, which was still in progress at the time [36]. An important aspect of reliably identifying compounds that have true positive interactions with a protein target is establishing ligand efficiency (LE) scores with known binding compounds. LE is a way to assess binding affinity that is normalized for the number of heavy atoms or non-hydrogen atoms in a compound [57]. The application of LE reference scores in large-scale screening and in vitro studies can greatly increase the likelihood of identifying true hits. A small, in-house library of 17 phosphorous-containing compounds was tested both in vitro and in silico to establish LE reference scores (Appendix A). 1D ^31^P NMR spectra were collected for each individual compound and in combination with DNAJA1-107 at a ligand:DNAJA1-107 ratio of 10:1. The compounds that demonstrated line broadening in the 1D ^31^P NMR spectra were then analyzed with a 2D ^1^H-^15^N HSQC NMR spectra. Chemical shift perturbations (CSPs) were mapped onto the DNAJA1-107 structure to identify the likely ligand binding site and differentiate between specific and non-specific binding to the DNAJA1 J-domain. CSPs were manually identified based on the peak center being visibly shifted or a decrease in peak intensity as evident by a change in the number of contours. A total of four compounds, ADP, O-phospho-L-serine, O-phospho-L-tyrosine, and dihydroxyacetone phosphate exhibited specific binding to DNAJA1-107 (Figure 2A,B). These compounds, their theoretical binding scores, and the perturbed amino acid residues are detailed in Table 3.

It is important to note that the LE scores for these four compounds had a wide range of values, including three negative and one positive value. Negative LE scores are generally indicative of a compound that may be a potential binder. These results provide evidence of the true arbitrary nature of scoring functions used in virtual screens and emphasize the necessity of reference compounds to facilitate successful hit-to-lead endeavors.

After reference LE scores were established, a larger virtual screen was completed using the Chembridge™ Diversity Library which contains 100,000 compounds and a total of 500,000 poses. Overall, the lowest and highest observed LE scores were −8.54 and 6.29, respectively. Approximately 3000 poses out of 500,000 had an LE score consistent with the four reference LE scores that were established using the phosphorous library. Only 23 compounds had an LE score higher than the LE reference score of 1.76. Only 20 compounds out of the 3000 poses consistent with the reference LE scores and from the entire 100,000-compound library were selected for further analysis. Specifically, five compounds were randomly selected from each of the four LE reference scores. This subset of 20 compounds were then screened by NMR to confirm binding by using a multi-step approach consisting of an STD experiment followed by a 2D ^1^H-^15^N HSQC experiment (Figure 2C,D) [58]. Five of the 20 compounds tested demonstrated binding to DNAJA1-107 by NMR. The list of compound IDs and associated in silico and in vitro data are provided in Table 4.

The compound structures, ChemBridge™ IDs, and IUPAC names can be found in Appendix A. Tanimoto structure similarity scores were calculated between all pairs of structures and yielded a high score of 0.41 and a low score of 0.16 indicating a relatively low compound similarity. The complete list of Tanimoto similarity scores is provided in Appendix A. ITC was then used to measure the dissociation constants for the DNAJA1-107 binders. Compounds 9101204 and 7727968 yielded dissociation constants (K_D_s) of 0.1 ± 0.3 µM and 10 ± 85 µM, respectively. ITC data plots can be found in Appendix A. The chemical shift perturbations in the 2D ^1^H-^15^N HSQC experiment (Figure 2C,D) appear relatively modest compared to these ITC K_D_ values. The fact that the ITC data also identifies multiple binding sites may partly explain the small CSPs. There may be fast off-rates as the ligand rapidly exchanges between these multiple sites. In addition to the K_D_ value, the magnitude of the CSPs is also determined by the type of interactions that are present with different interactions yielding highly variable CSPs (i.e., salt-bridges, hydrogen-bonds, ring stacking, van der Waal interactions, etc.). Thus, a combination of a short binding site lifetime and interactions that yield small CSPs may explain the apparent differences between the ITC, STD, and HSQC results.

Due to poor solubility in DMSO, the remaining three compounds could not be analyzed by ITC. It is again worth noting that of the five compounds confirmed to bind DNAJA1-107 from the large virtual library, two had positive LE scores and three had negative LE scores. As positive LE values are generally considered to be indicative of poor binding potential, the value of establishing the LE reference scores before screening in vitro was paramount to successful hit identification.

### 3.4. Identification of Ligand Binding Sites on DNAJA1-107

A surface structure depiction of DNAJA1-107 highlights the residues exhibiting ligand-induced CSPs in the 2D ^1^H-^15^N HSQC spectra (Figure 3A,B). Two distinct ligand binding sites were apparent with one centered around the HPD motif in the J-domain, and the other ligand binding site located at the end of α-helix 4. The HPD motif is a highly conserved, three amino-acid sequence that is located between the second and third α-helix in the J-domain. This motif is a defining feature of all J-domains and is essential for the interaction of Hsp40s with Hsp70s [59]. Previous studies have also demonstrated mutations of the HPD motif can completely eliminate the function of the J-domain, which further supports the functional importance of this motif to DNAJA1 [60]. The second ligand binding site located after α-helix 4 also supports the value of the linker region to DNAJA1 biological activity. The glycine-rich linker may facilitate the binding of ligands to a protein domain or domains; and as demonstrated by our in silico and in vitro studies, is likely vital to the function of DNAJA1 in vitro [54]. The propensity of two binding sites on DNAJA1-107 is also supported by the ITC data collected for compounds 9101204 and 7727968 that had molar ratios of 2.35 and 2.54, respectively. The identified HPD and linker ligand binding regions are both hydrophilic, as would be anticipated from solvent-exposed domains that play vital roles in protein interactions. As was previously noted, the HPD binding region contains a positively charged surface at the end of α-helix 2, which may serve as a binding location for a negatively charged surface of Hsp70s [4]. However, the binding region at the end of α-helix 4 is negatively charged, which suggests a role outside of facilitating Hsp70 interactions (Figure 3C,D).

### 3.5. DNAJA1-107 Protein Binding Partners

A standard pull-down assay was completed to identify potential protein binding partners of DNAJA1. The assay was done in vitro utilizing a protein extract from the PDAC cell line MIA PaCa-2 and a recombinant DNAJA1-107. The recombinantly expressed DNAJA1-107 with a 6x-histidine affinity tag was immobilized on a HisPur™ cobalt resin followed by the addition of the protein cell extract. A negative control used the cobalt resin without the immobilized protein to identify non-specific binding interactions. The protein-resin or resin-only mixtures were incubated for 2 h to provide sufficient time to facilitate protein–protein or non-specific binding interactions. The bound DNAJA1-protein samples were co-eluted and resolved by SDS-PAGE. Gel bands were excised, trypsin digested, and identified by tandem MS. A total of eight human proteins were identified with four having nuclear subcellular locations and four having mitochondrial subcellular locations. None of these proteins were observed to bind to the resin-only mixture suggesting a specific interaction with DNAJA1. A full list of identified proteins can be found in Table 5. These proteins are consistent with the cellular localization of DNAJA1 in the cytosol and around the nucleus [7]. While p53 was not identified in our assay, DNAJA1 has been proposed to interact with p53, [25,35,61] which may account for the identification of mitochondrial proteins in the pull-down assay. Our results are also supported by DnaJ family member DNAJA3, which is proposed to be a mediator of p53-induced apoptosis by promoting mitochondrial localization of p53 [62].

Recombinant human protein ADP/ATP translocase 3 (ANT3) was obtained from CUSABIO to confirm an in vitro interaction of DNAJA1. ANT3 is a transmembrane protein that mediates the import of ADP and the export of ATP in the mitochondrial matrix. Decreased expression of ANT3 has been implicated in resistance to TNF-α-induced cell death and the increased production and resistance to reactive oxygen species (ROS) [63]. Decreased cell death and increased production and resistance to ROS were observed in our previous cellular studies where DNAJA1 was overexpressed, [6] suggesting there may be a link between the role of DNAJA1 and ANT3 that promotes cellular survival. ANT3 was supplied in a 10 mM tris-HCl buffer with 1 mM EDTA and 6% trehalose at pH 8.0. ^15^N-labeled DNAJA1-107 was confirmed for stability and proper folding in this buffer. Immediate protein precipitation was apparent when the two proteins were combined at an equal molarity of 50 µM. ANT3 was then decreased to 1/10^th^ the concentration of DNAJA1-107 in an attempt to prevent protein precipitation. Unfortunately, protein precipitation again occurred immediately upon combining the two proteins. Numerous other conditions were tested, including increasing NaCl and CaCl_2_ concentrations, reversing titration order, decreasing the concentrations of the proteins, and switching to an MES buffer at a lower pH. Unfortunately, none of these conditions improved the outcome, or prevented complete protein precipitation. A titration of DNAJA1-107:ANT3 at molar ratios of 10:1, 5:1, and 1:1 did demonstrate the protein precipitation was concentration dependent. At higher DNAJA1-107 to ANT3 ratios, 2D ^1^H-^15^N HSQC peaks for DNAJA1-107 were still detectable, which suggests the interaction of DNAJA1-107 with ANT3 leads to its precipitation while the unbound DNAJA1-107 remains correctly folded and in solution. While not an ideal or preferred outcome, our results clearly indicate a protein–protein interaction between DNAJA1-107 and ANT3 did occur. Stabilizing the DNAJA1-107:ANT3 complex may require any number of preferred conditions, such as the full DNAJA1 construct, significantly lower concentrations, or the presence of a membrane.

## 4. Conclusions

The NMR solution structure of the J-domain with the glycine- and phenylalanine-rich linker region was presented, which confirmed a primarily unstructured motif with some α-helical propensity that have been previously shown to aid the in vivo functionality of DNAJA1 [56]. It was surprising to observe that the addition of the linker region to the J-domain resulted in both a structurally disordered linker region and a disruption in the helical structure of the J-domain. The high Gly-ratio of the linker region may explain the observed decrease in a defined structure. Virtual and in vitro screens were able to identify an additional ligand binding site on DNAJA1-107 within the glycine-rich linker region. The two ligand binding sites on DNAJA1-107 were supported by the ITC data and NMR CSPs. Our results support the importance of the linker region in facilitating protein: ligand interactions. Our previous metabolomics study investigating the impact of overexpressing DNAJA1 in PDAC cell lines suggested a potential proto-oncogenic role for DNAJA1. This is further supported by the results of the pull-down assay where three (ANT3, glutamate dehydrogenase, and creatine kinase) of the potential protein binding partners of DNAJA1 have been implicated in cancer progression. A decrease in the expression of ANT3 leads to an anti-apoptotic state and increased ROS resistance [63]. Alternatively, glutamate dehydrogenase upregulation in colorectal cancer promotes cellular proliferation, migration, and invasion in vitro [64] and the upregulation of creatine kinase regulates cell cycle progression and can promote cell division in breast cancer [61]. The success of the virtual ligand screens was dependent on establishing LE scores with known DNAJA1 binders. The results of the virtual ligand screens and pull-down assay suggests the linker region is vital to in vitro studies of DNAJA1 functionality. DNAJA1 may promote oncogenesis by inhibiting or stimulating other proteins necessary for cellular growth and survival. While our results demonstrate the potential importance of DNAJA1 interactions with proteins and ligands to cancer, additional in vitro studies are required to confirm these protein interactions, and to validate that our chemical leads successfully compete with these DNAJA1:protein interactions and exhibit biological activity.

## Figures and Tables

**Figure 1 biomolecules-12-01391-f001:**
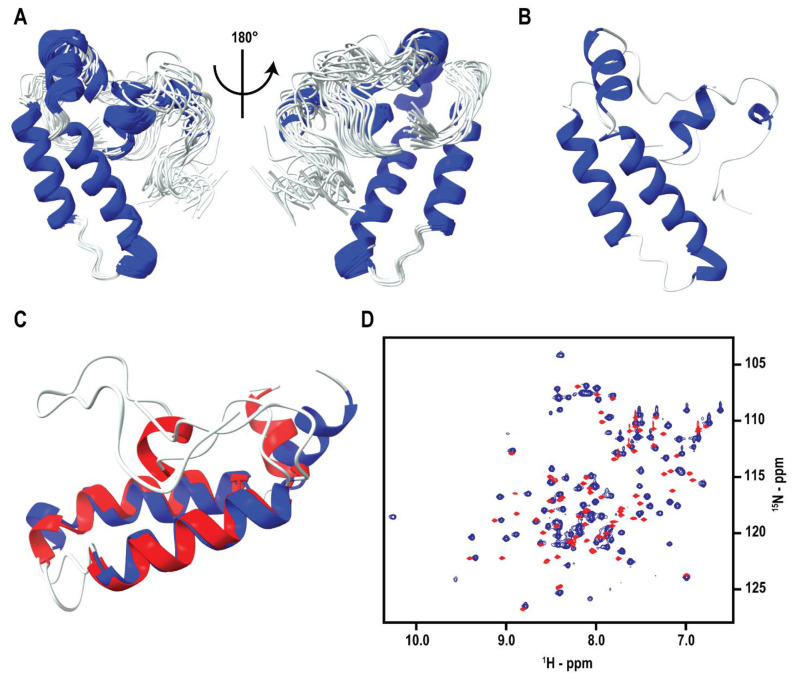
DNAJA1-107 Structural Comparison. (**A**) Ribbon overlay of the 20 lowest-energy, water-refined NMR structures of DNAJA1-107. (**B**) Ribbon representation of the average DNAJA1-107 structure from the water-refined ensemble in (**A**), which was further water-refined. Navy represents α-helices and white is unstructured loops. (**C**) Ribbon overlay of the J-domains of DNAJA1-67 (red) and DNAJA1-107 (navy). (**D**) Overlay of the 2D ^1^H-^15^N HSQC spectra of DNAJA1-67 (red) and DNAJA1-107 (navy).

**Figure 2 biomolecules-12-01391-f002:**
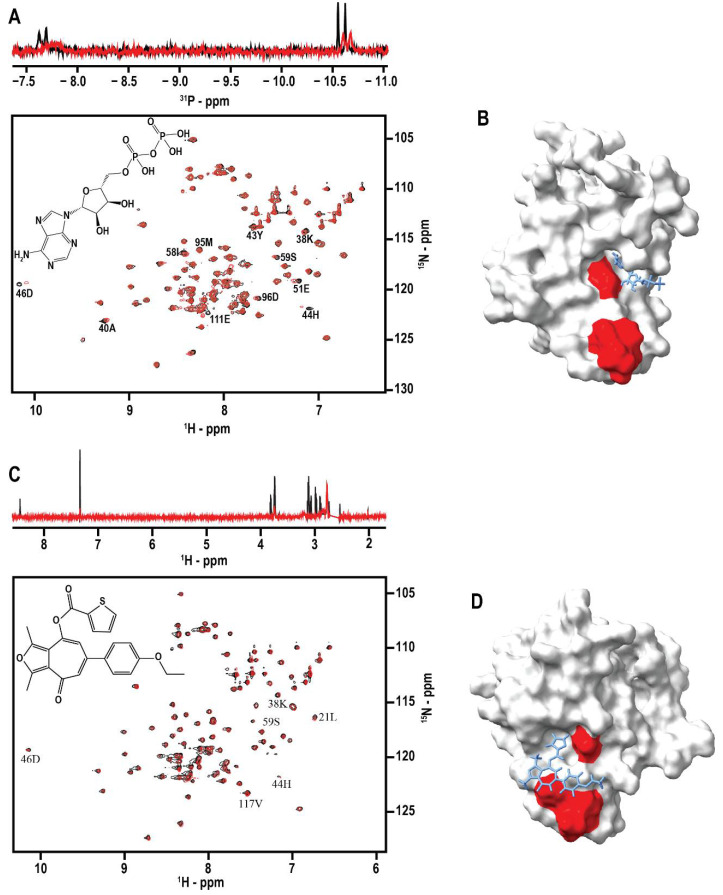
NMR Evaluation of Ligand Binding to DNAJA1-107. (**A**) (top) Overlay of 1D ^31^P spectra of ADP in the presence (red) and absence (black) of DNAJA1. The significant line-broadening of the ADP phosphorus NMR resonances is indicative of protein binding. (bottom) Overlay of the 2D ^1^H-^15^N HSQC spectra of apo-DNAJA1-107 (black) and DNAJA1-107 bound to ADP (red). (**B**) Surface rendition of the DNAJA1-107 NMR solution structure with residues experiencing an ADP-induced CSP colored red. ADP is shown as a licorice drawing in its MVD predicted binding pocket. (**C**) (top) Overlay of the 1D STD spectra of compound 7727968 in the presence (red) and absence (black) of DNAJA1-107. The observation of NMR resonances only in the STD experiment in the presence of DNAJA1 is evidence of compound 7727968 binding DNAJA1. The residual DMSO solvent peak at 2.8 ppm was removed from the spectra for clarity. (bottom) Overlay of the 2D ^1^H-^15^N HSQC spectra of apo-DNAJA1-107 (black) and DNAJA1-107 bound to 7727968 (red). (**D**) Surface rendition of the DNAJA1-107 NMR solution structure with residues experiencing a 7727968-induced CSP colored red. 7727968 is shown as a licorice drawing in its MVD predicted binding pocket.

**Figure 3 biomolecules-12-01391-f003:**
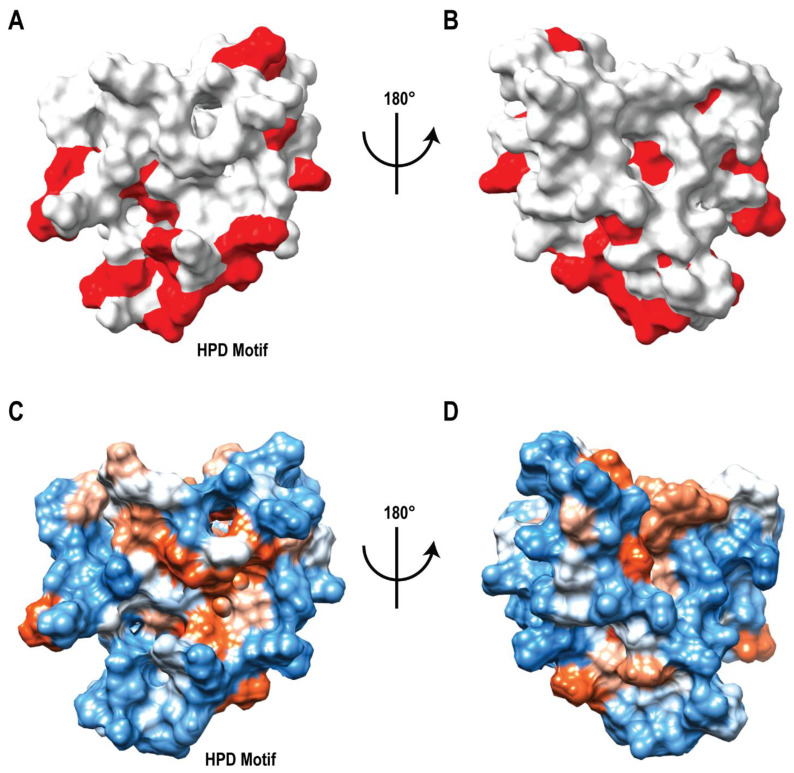
DNAJA1-107 Ligand Binding Sites. (**A**) Front and (**B**) back views of the surface rendition of the DNAJA1-107 NMR solution structure with ligand interaction sites confirmed by CSPs colored red. (**C**) Front and (**D**) back views of the surface rendition of the DNAJA1-107 NMR solution structure colored by hydrophobicity. Blue is hydrophilic and red is hydrophobic.

**Table 1 biomolecules-12-01391-t001:** Statistics for the DNAJA1-107 NMR Structure.

	<SA> ^a^	${\left(\bar{\mathrm{SA}}\right)}_{r\;}$ ^b^
RMSD for distance restraints (experimental) (Å)		
All (1143)	0.094 ± 0.006	0.078
Inter-residue sequential (|i − j| = 1) (260)	0.066 ± 0.005	0.069
Inter-residue short-range (1 < |i − j| < 5) (274)	0.079 ± 0.007	0.084
Inter-residue long-range (|i − j| ≥ 5) (84)	0.098 ± 0.005	0.104
Intraresidue (525)	0.108 ± 0.009	0.079
H-bonds (23)	0.015 ± 0.003	0.010
RMSD for dihedral angle restraints (deg) (176)	0.844 ± 0.117	1.068
RMSD for ^3^J_NHα_ restraints (Hz) (39)	0.897 ± 0.047	0.923
RMSD (covalent geometry)		
Bonds (Å)	0.007 ± 0.000	0.007
Angles (deg)	0.727 ± 0.019	0.742
Impropers (deg)	1.072 ± 0.020	1.060
energy (kcal/mol)		
Total	−3131.62 ± 104.82	−3446.54
Bonds	38.70 ± 2.50	39.70
Angles	160.06 ± 9.35	162.09
Dihedrals	8.73 ± 2.24	7.43
Impropers	66.45 ± 6.32	68.53
Van der Waals	−241.02 ± 11.94	−216.43
NOE	108.92 ± 11.60	122.25
3JNHα	30.73 ± 11.60	33.20

^a^ <SA> represents the final 20 water-refined simulated annealing structures. ^b^$\;{\left(\bar{SA}\right)}_{r}\;$ represents the water-refined average structure of all 20 water-refined structures.

**Table 2 biomolecules-12-01391-t002:** Evaluation of the DNAJA1-107 NMR Structure.

PSVS Z-Score (Ordered Residues ^a^)	
Verify3D	−2.25
ProsaII (-ve)	0.83
Procheck (φ and ψ)	0.63
Procheck (all)	0.18
MolProbity clash score	−2.64
Ramachandran space (all residues)	
Most favored regions	90.5%
Additionally allowed regions	9.5%
Disallowed regions	0%

^a^ residues 2–57, 59–63, 66–86, 88–95.

**Table 3 biomolecules-12-01391-t003:** Phosphorous compounds that bind DNAJA1-107.

Compound Name	LE Score	1D Hits ^a^	2D Hits ^b^	Perturbed Residues
Adenosine 5′-diphosphate	−3.32	X	X	38K, 40A, 43Y, 44H, 46D, 51E, 58I, 59S, 95M, 96D, 111E
O-phospho-L-serine	−1.73	X	X	38K, 40A, 43Y, 44H, 46D, 51E, 58I, 59S, 77G, 80Q, 93S, 96D, 111E
Dihydroxyacetone phosphate	−1.53	X	X	38K, 44H, 46D, 51E, 96D, 111E
O-phospho-L-Tyrosine	1.76	X	X	30E, 38K, 40A, 41L, 43Y, 44H, 46D, 50N, 52G, 58I, 59S, 77G, 93S, 103G, 111E

^a^ Compounds that demonstrated line broadening in 1D ^31^P NMR spectra. ^b^ Compounds that demonstrated CSPs in 2D ^1^H-^15^N HSQC spectra.

**Table 4 biomolecules-12-01391-t004:** Hit2Lead compounds that bind DNAJA1-107.

Hit2Lead ID	Compound Name	1D Hits ^a^	2D Hits ^b^	LE Score	Perturbed Residues	K_D_ (μM) ^c^
7727968	6-(4-ethoxyphenyl)-1,3-dimethyl-4-oxo-4H-cyclohepta[c]furan-8-yl 2-thiophenecarboxylate	X	X	1.76	117V, 21L, 38K, 44H, 46D, 59S	10 ± 85
7912207	N-{5-[(diethylamino)sulfonyl]-2-methoxyphenyl}-2-methoxybenzamide	X	X	1.76	40A, 44H, 46D, 51E, 58I, 59S, 93S, 96D	N.D.
9042610	N-(5-chloro-2-methoxyphenyl)-2-[4-(3-methoxyphenyl)-1-piperazinyl]acetamide	X	X	−1.73	38K, 40A, 43Y, 44H, 46D, 51E, 58I	N.D.
9080132	N-(3-methylphenyl)-4-(5-methyl-1H-1,2,3-triazol-1-yl)benzamide	X	X	−3.32	38K, 40A, 44H, 46D, 51E, 58I, 93S, 96D	N.D.
9101204	1-[(5-bromo-2-thienyl)carbonyl]-4-(4-fluorophenyl)piperazine	X	X	−3.32	27A, 38K, 46D, 58I, 59S, 79E, 96D, 117V	0.1 ± 0.3

^a^ Compounds that demonstrated a positive STD NMR spectrum. ^b^ Compounds that demonstrated CSPs in 2D ^1^H-^15^N HSQC spectra, ^c^ The ITC data (Appendix A) were fit to a single K_D_ value using a standard binding curve with stoichiometry set as a variable. N.D.–no data.

**Table 5 biomolecules-12-01391-t005:** Proteins identified in pull-down assay.

UniProt ID	Gene	Protein	Localization	Matched Peptides	Sequence Coverage (%)
P12236	SLC25A6	ADP/ATP translocase 3	mitochondria	1	10.1
P23246	SFPQ	Splicing factor, proline- and glutamine-rich	nucleus	22	30
P14136	GFAP	Glial fibrillary acidic protein	cytoplasm	4	12.5
P02545	LMNA	Prelamin-A/C	nucleus	34	52.7
Q15233	NONO	Non-POU domain-containing octamer-binding protein	nucleus	11	20.4
P00367	GLUD1	Glutamate dehydrogenase 1, mitochondrial	mitochondria	18	28.7
Q8WV22	NSMCE1	Non-structural maintenance of chromosomes element 1 homolog	nucleus	6	12.4
P12532	CKMT1A	Creatine kinase U-type, mitochondrial	mitochondria	9	22.1

## Data Availability

Chemical shift assignments have been deposited into the BMRB with ID 51532. The coordinates of the water-refined ensemble have been deposited into the PDB with ID 8E2O.

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
