# Peer review of "Leveraging the Structure of DNAJA1 to Discover Novel Potential Pancreatic Cancer Therapies"

_biomolecules, 2022, doi:10.3390/biom12101391_

Round 1
Reviewer 1 Report
The article describes a computationally assisted NMR screen of drug-like compounds that bind to DNAJA1, a protein that might be important in future cancer therapy. In addition, the NMR structure of DNAJA1 is reported. The manuscript overall is very clear and well presented. I have only minor criticism/suggestions.
1. I recommend toning down the title since the link to therapeutic development is still tenuous. The language used elsewhere, such as drug-like molecules that bind to DNAJA1 is more clearly what is reported.
2. I believe the first sentence is technically incorrect. The 62,219 diagnoses are just those in the USA.
3. In the materials and methods it would help to indicate the yield of bio-synthesized protein per liter.
4. What was the experiment that yielded 3JNHa couplings? In the methods, only assignment experiments are mentioned.
5. section 3.3, what was the cutoff (or other criteria) used to identify whether or not there were significant CSPs?
6. Fig 2. is missing explanation of the 1D spectra. Also, please indicate the reason for the missing/cut out region at 2.8 ppm
7. line 330 there is a statement that 5 compounds were selected for each LE ref. score. Out of how many? i.e. how many scored better than the worst LE ref. score of 1.76? (how many of the hits remain unscreened by NMR?).
8. line 393, the text states to confirm (more than one of the) partners, but only one was tested. This should be stated more carefully. Also, what is done to control for non-specific binding to the cobalt column resin?
some typos:
line 76 'been shown'
210 loop regions
344 _of_ the five
446- validate _that_
Author Response
- The article describes a computationally assisted NMR screen of drug-like compounds that bind to DNAJA1, a protein that might be important in future cancer therapy. In addition, the NMR structure of DNAJA1 is reported. The manuscript overall is very clear and well presented.
Response: We appreciate the Reviewers comments, which have contributed to an improved manuscript.
- I recommend toning down the title since the link to therapeutic development is still tenuous. The language used elsewhere, such as drug-like molecules that bind to DNAJA1 is more clearly what is reported.
Response: To address the Reviewer’s concerns, we have added the word “Potentially” to the title: “Leveraging the Structure of DNAJA1 to Discover Potentially Novel Pancreatic Cancer Therapies”. Since both our own prior work (PMC8823314, PMC3985919) and the scientific literature (PMC7948493, PMC8804903) have established DNAJA1 as important to cancer in general and to pancreatic cancer in particular, we believe it is relevant and important to include a reference to pancreatic cancer in the title of our paper.
- I believe the first sentence is technically incorrect. The 62,219 diagnoses are just those in the USA.
Response: The Reviewer is correct. The sentence was revised to clearly indicate people in the USA.
- In the materials and methods it would help to indicate the yield of bio-synthesized protein per liter.
Response: The approximate yield of purified DNAJA1-107 was 8-10 mg/L in M9 media. This statement was added to the methods.
- What was the experiment that yielded 3JNHa couplings? In the methods, only assignment experiments are mentioned.
Response: The 3JNHa couplings were measured from the HNHA experiment, which was inadvertently left off the list of NMR experiments in section 2.4. This omission was corrected in the revised manuscript.
- section 3.3, what was the cutoff (or other criteria) used to identify whether or not there were significant CSPs?
Response: A chemical shift changed was identified by visual inspection. If the center of the peak was visibly shifted or if the peak intensity was decreased as evident by differences in the number of contours. This definition was added to the revised manuscript in section 3.3.
- Fig 2. is missing explanation of the 1D spectra. Also, please indicate the reason for the missing/cut out region at 2.8 ppm
Response: This was an unfortunate omission. The legend to figure 2 has been corrected to include a description of the 1D 31P line broadening (Figure 2A) and STD (Figure 2C) spectra. The region of the spectra around 2.8 ppm corresponded to the residual DMSO signal that was removed for clarity. This point is now clearly stated in the figure legend in the revised manuscript.
- line 330 there is a statement that 5 compounds were selected for each LE ref. score. Out of how many? i.e. how many scored better than the worst LE ref. score of 1.76? (how many of the hits remain unscreened by NMR?).
Response: Approximately 3000 poses out of 500,000 had an LE score consistent with the four reference LE scores. Only 23 compounds had an LE higher that the LE reference score of 1.76. Only 20 compounds out of the 3000 poses consistent with the reference LE or the 100,000 compounds in total were screened by NMR. These points were clarified in the revised text in sections 2.2 and 3.3.
- line 393, the text states to confirm (more than one of the) partners, but only one was tested. This should be stated more carefully.
Response: We agree, the original sentence created an unintentional misrepresentation of our study: “To confirm the in vitro interaction of DNAJA1 with these potential binding partners, recombinant human protein ADP/ATP translocase 3 (ANT3) was obtained from CUSABIO”. In the revised manuscript, the sentence was simplified to: “Recombinant human protein ADP/ATP translocase 3 (ANT3) was obtained from CUSABIO to confirm an in vitro interaction of DNAJA1.”
- Also, what is done to control for non-specific binding to the cobalt column resin?
Response: A negative control consisted of incubating a cell lysate with the cobalt-only resin. None of the proteins retained to the cobalt resin with the immobilized DNAJA1 were detected in the cobalt-only resin. Sections 2.5.2 and 3.5 were revised to include a description of the control experiment.
- some typos:
line 76 'been shown'
210 loop regions
344 _of_ the five
446- validate _that_
Response: All the typos identified by the Reviewer have been corrected in the revised manuscript.
Reviewer 2 Report
Peer review of: “Leveraging the Structure of DNAJA1 to Discover Novel Pancreatic Cancer Therapies” by Roth et al. for Biomolecules.
The authors present an NMR structure in explicit solvent of 107 amino acid domain of the cancer target chaperone component DNAJa1; prior structure was only 67 amino acids and so this new structure could provide a more accurate model for drug discovery. In addition, they performed virtual screening of a large library of 100,000 structures) and found 5 binders that were confirmed by NMR and ITC. Tacked on is a study with in vitro pull downs to finding binding partners to DNAJa1.
The Good - Large library screening with some validation of reference compounds. Structure calculation was performed well, particularly with inclusion of refinement in explicit solvent using RECCORD parameterization. Ligand docking methodology appears appropriate with some hits that could be validated experimentally. Two of those hits bind quite tightly, low micromolar to high nanomolar Kd and are promising hits for further chemical elaboration.
The Bad – The impact of the NMR structure upon the rest of the study is minimal since they performed their virtual screening on a prior homology model. Essentially the NMR work was just used to generate resonance assignments and as a model for interpreting CSP validation. The magnitude of CSPs for compound 7727968 seem astonishing small for such tight binder; authors should address this. The significance of the 1D spectra above 2D HSQCs in figure 2 are not described; Are these STD or T2 on compounds? I assume latter since broadening of reference set described in text. Please update Fig. 2 caption to clarify what those spectra represent. The authors try to make a case that they can identify general binding sites on DNAJa1, i.e. figure 3, but the limited number of CSP validated hits may not be enough to support this notion. The pull-down studies and study with Ant3 (somewhat ambiguous/negative result really) feel tacked on and don’t complement the ligand studies and may be considered to be published in a separate work.
The Ugly – Buried the lede. There is an annoyingly long review of the significance of pancreatic cancer and review of current pharmaceuticals at beginning of introduction that detracts from impact of the study. Rather than providing so much superfluous background, lean on the 2020 study from Truman’s group that identified DNAJA1 as putative cancer target from chemogenomic screening. That work succinctly justifies studying DNAJa1 as cancer target. The last paragraph, lines 89 to 99 should be front forward in the introduction; authors will not want to read the lengthy background which is presumably taken whole cloth from Powers’ NIH grant proposal(s).
Author Response
- The Good - Large library screening with some validation of reference compounds. Structure calculation was performed well, particularly with inclusion of refinement in explicit solvent using RECCORD parameterization. Ligand docking methodology appears appropriate with some hits that could be validated experimentally. Two of those hits bind quite tightly, low micromolar to high nanomolar Kd and are promising hits for further chemical elaboration
Response: We appreciate the Reviewer’s support for our paper and its findings.
- The Bad – The impact of the NMR structure upon the rest of the study is minimal since they performed their virtual screening on a prior homology model. Essentially the NMR work was just used to generate resonance assignments and as a model for interpreting CSP validation.
Response: We respectfully disagree with the reviewer’s assertion that this outcome is “bad”. Instead, we believe the results are very positive and encouraging. It demonstrates both the robustness of the virtual screen, the NMR structural work and the NMR/ITC ligand screen. Simply, all the results are internally consistent and yielded the same outcome. Conversely, relying exclusively on computational modeling without experimental verification is fraught with problems and likely false leads.
- The magnitude of CSPs for compound 7727968 seem astonishing small for such tight binder; authors should address this.
Response: We agree, and a possible explanation was added to section 3.3. Briefly, the presence of multiple binding sites, an exchange between these sites, and the type of ligand-protein interaction may combine to produce modest CSPs.
- The significance of the 1D spectra above 2D HSQCs in figure 2 are not described; Are these STD or T2 on compounds? I assume latter since broadening of reference set described in text.
Response: This was an unfortunate omission and clearly a point of confusion that was addressed in the revised manuscript. The legend to Figure 2 has been corrected to include a description of the 1D 31P line broadening (Figure 2A) and the STD (Figure 2C) spectra. Similarly, the text in sections 2.3.1 has been corrected to clarify that 1D 31P line broadening experiment was used to identify DNAJA1 binding from the in-house library of 17 phosphorus containing compounds. The STD experiment was used to identify binding to DNAJA1 by compounds identified from the virtual screen using the ChemBridge™ Diversity Library.
- Please update Fig. 2 caption to clarify what those spectra represent.
Response: The legend for Figure 2 has been updated to include a description of the 1D NMR spectra.
- The authors try to make a case that they can identify general binding sites on DNAJa1, i.e. figure 3, but the limited number of CSP validated hits may not be enough to support this notion.
Response: We disagree, and the Reviewer’s perspective may result from a misinterpretation of the results presented in Figure 2. The CSPs were simply overlayed onto the docked model predicted by Molegro Virtual Docker demonstrating the consistency between the experimental NMR data and the docked ligand structure on DNAJA1. Figure 3 then represents a consensus of the ligand induced CSPs (Tables 3 & 4) consistent with the docked structures.
- The pull-down studies and study with Ant3 (somewhat ambiguous/negative result really) feel tacked on and don’t complement the ligand studies and may be considered to be published in a separate work.
Response: We respectively disagree with the Reviewer. The results are not negative since there is clearly an interaction between DNAJA1 and the proteins from the pull-down assay (Table 3). The outcome does support the proposal that DNAJA1 may function in various cancers by selectively interacting and stabilizing mutated proteins like p53. While we do agree that these results are preliminary, but our expectation is that it may guide future studies. Importantly, the results do not stand-on its own and publishing the data in an alternative paper is not an option, and importantly, not publishing the results is not an acceptable alternative option.
- The Ugly – Buried the lead. There is an annoyingly long review of the significance of pancreatic cancer and review of current pharmaceuticals at beginning of introduction that detracts from impact of the study. Rather than providing so much superfluous background, lean on the 2020 study from Truman’s group that identified DNAJA1 as putative cancer target from chemogenomic screening. That work succinctly justifies studying DNAJa1 as cancer target.
Response: We agree with the Reviewer and the summary of current treatments for pancreatic cancer were deleted from the introduction.
- The last paragraph, lines 89 to 99 should be front forward in the introduction; authors will not want to read the lengthy background which is presumably taken whole cloth from Powers’ NIH grant proposal(s).
Response: We agree with the Reviewer and the last paragraph was added to the first paragraph of the introduction.
Reviewer 3 Report
In this paper, Powers and co-workers report the NMR structure of the first ~100 residues of the DNAJA1, a protein involved in the progression of pancreatic cancer. The canonical J domain constitutes the N-terminal domain of DNAJA1, with relatively compact helical domains. The authors utilized typical heteronuclear NMR experiments to determine the three-dimensional structure. The calculations result in a well-defined ensemble of conformers with a standard resolution found for NMR structures deposited in the PDB. The authors utilized the NMR structure to screen for possible ligands and define sites that drugs can target. The virtual screening results in a subset of compounds. These researchers screened a subset of compounds using NMR and ITC.
Overall, the paper is interesting and provides initial information for future drug discovery. The paper will be suitable for publication in your journal after the authors address the following points:
A) The authors must provide a table with the experimental dissociation constants for the compounds analyzed.
B) What is the exchange regime for the compounds tested? Did the author perform titrations for the ligands, or did they analyze only 10:1 ratios? The authors need to provide NMR titration data.
C) The ITC data are puzzling. What model was used to fit data in Figure S3? The binding shows two separate events, and the model utilized to fit the data seems to be inadequate to represent the experimental trend. Figure 4 is also problematic. The observed behavior is not defined by the model used. The authors should provide the experimental ITC profiles (heat release vs. ligand concentration).
Author Response
- Overall, the paper is interesting and provides initial information for future drug discovery. The paper will be suitable for publication in your journal after the authors address the following points.
Response: We appreciate the Reviewer’s support for our paper and its findings.
- The authors must provide a table with the experimental dissociation constants for the compounds analyzed.
Response: The two measured KDs were added to Table 4.
- What is the exchange regime for the compounds tested?
Response: Since we observe chemical shift changes, a single set of peaks for both ligands and protein, and some peak broadening, we are likely in the fast to intermediate exchange regime.
- Did the author perform titrations for the ligands, or did they analyze only 10:1 ratio? The authors need to provide NMR titration data.
Response: We only performed single-point NMR titrations and relied on the ITC data to provide estimates of dissociation constants.
- The ITC data are puzzling. What model was used to fit data in Figure S3? The binding shows two separate events, and the model utilized to fit the data seems to be inadequate to represent the experimental trend. Figure 4 is also problematic. The observed behavior is not defined by the model used. The authors should provide the experimental ITC profiles (heat release vs. ligand concentration).
Response: The ITC data in Figures S3 and S4 have been updated to include the raw data. The raw data was fit to a standard binding curve with stoichiometry set as a variable.